# Detection of Tahyna Orthobunyavirus-Neutralizing Antibodies in Patients with Neuroinvasive Disease in Croatia

**DOI:** 10.3390/microorganisms10071443

**Published:** 2022-07-18

**Authors:** Tatjana Vilibic-Cavlek, Vladimir Stevanovic, Vladimir Savic, Domagoj Markelic, Dario Sabadi, Maja Bogdanic, Snjezana Kovac, Marija Santini, Irena Tabain, Tanja Potocnik-Hunjadi, Ivana Ferencak, Ana Marija Skoda, Ana Sankovic, Ljubo Barbic

**Affiliations:** 1Department of Virology, Croatian Institute of Public Health, 10000 Zagreb, Croatia; domagoj.markelic@student.mef.hr (D.M.); maja.bogdanic@hzjz.hr (M.B.); irena.tabain@hzjz.hr (I.T.); ivana.ferencak@hzjz.hr (I.F.); amskoda1@gmail.com (A.M.S.); 2School of Medicine, University of Zagreb, 10000 Zagreb, Croatia; marijasantini.ms@gmail.com (M.S.); asankovic@outlook.com (A.S.); 3Department of Microbiology and Infectious Diseases with Clinic, Faculty of Veterinary Medicine, University of Zagreb, 10000 Zagreb, Croatia; snjezana.kovac@vef.unizg.hr (S.K.); ljubo.barbic@vef.unizg.hr (L.B.); 4Poultry Center, Croatian Veterinary Institute, 10000 Zagreb, Croatia; v_savic@veinst.hr; 5Department of Infectious Diseases, Clinical Hospital Center Osijek, 31000 Osijek, Croatia; dariocroatia@gmail.com; 6Medical Faculty, Josip Juraj Strossmayer University of Osijek, 31000 Osijek, Croatia; 7Department for Adult Intensive Care and Neuroinfections, University Hospital for Infectious Diseases “Dr Fran Mihaljevic”, 10000 Zagreb, Croatia; 8Department of Infectious Diseases, General Hospital Varazdin, 42000 Varazdin, Croatia; tanja.potocnik.h@gmail.com

**Keywords:** Tahyna orthobunyavirus, neuroinvasive disease, neutralizing antibodies, Croatia

## Abstract

Background: Tahyna orthobunyavirus (TAHV) is widely distributed in continental Europe. Very few studies have analyzed TAHV seroprevalence in Croatia. We analyzed the prevalence of TAHV RNA and antibodies in Croatian patients with neuroinvasive disease (NID). Methods: A total of 218 patients with unsolved NID detected during five consecutive arbovirus transmission seasons (April 2017–October 2021) were tested. Cerebrospinal fluid (CSF) and urine samples were tested for TAHV RNA using RT-PCR. In addition, CSF and serum samples were tested for TAHV antibodies using a virus neutralization test (VNT). Results: Clinical presentations in patients with NID were meningitis (141/64.7%), meningoencephalitis (56/25.7%), myelitis (8/3.7%), and ‘febrile headache’ (13/5.9%). TAHV RNA was not detected in any of the tested CSF or urine samples; however, TAHV-neutralizing (NT) antibodies were detected in 22/10.1% of patients. Detection of NT antibodies in the CSF of two patients presenting with meningitis suggested recent TAHV infection. TAHV seropositivity increased significantly with age, from 1.8% to 24.4%. There was no difference in seroprevalence between genders or areas of residence (urban, suburban/rural). The majority of seropositive patients (90.9%) resided in floodplains along the rivers in continental Croatia. Conclusions: The presented results confirm that TAHV is present in Croatia. The prevalence and clinical significance of TAHV infection in the Croatian population have yet to be determined.

## 1. Introduction

Tahyna orthobunyavirus (TAHV) is a mosquito-borne virus that belongs to the family *Peribunyaviridae*, genus *Orthobunyavirus*, California encephalitis serogroup. As with all bunyaviruses, TAHV has a tri-segmented negative-sense RNA genome with large (L), medium (M), and small (S) genome segments. The virus contains four structural proteins: two surface glycoproteins (Gn and Gc), the nucleocapsid protein (N), and the large (L) protein, an RNA-dependent RNA polymerase [1].

The virus was first isolated from mosquitoes near Tahyna village in Slovakia in 1958 [2], while the disease caused by TAHV (“Valtice fever”) was confirmed in the 1960s after the virus was isolated from humans with acute influenza-like illness in the Czech Republic, near Valtice (South Moravia). In the enzootic cycle, the vertebrate hosts for TAHV are hares, rabbits, hedgehogs, and rodents, while culicine mosquitoes, *Aedes vexans*, are considered the principal arthropod vectors [3]. Humans represent incidental or dead-end hosts in the transmission cycle. Natural foci of TAHV occur in flooded lowland habitats (floodplain forest ecosystem), sometimes including periurban areas [4].

TAHV is widely distributed in continental Europe, as shown by the detection and isolation of the virus from mosquitoes and by serological evidence from animals and humans [5,6,7]. Outside of Europe, TAHV infections have been reported in Asia (southern Siberia and the Far East, Turkey, Armenia, Azerbaijan, Tajikistan, Uzbekistan) and Africa (Tunisia, Morocco, Egypt, Ethiopia, Mozambique, Uganda, Kenya, Angola, South Africa, West Africa). In addition, TAHV antibodies have been found in Sri Lanka and China [6,8,9,10,11]. Furthermore, TAHV sequences were detected in two Chinese symptomatic patients in 2006 [12]. 

The majority of TAHV infections are asymptomatic [13]. Human disease caused by TAHV is an influenza-like illness occurring in late summer and early autumn, mainly in children [4]. Clinical symptoms are milder in adults than in children [14]. The disease usually presents as fever, gastrointestinal disorders, and occasionally atypical pneumonia or myocarditis. No fatal cases have been reported due to TAHV infection [15]. Despite its association with neurovirulence (meningitis), TAHV infection remains a neglected disease, with few reports of human clinical infections in recent decades [3]. However, neutralizing (NT) antibody seroprevalence rates of up to 60–80% have been regularly documented in adult human populations in endemic countries such as the Czech Republic [16].

Diagnosis of TAHV is confirmed by detection of TAHV RNA by reverse transcription-polymerase chain reaction (RT-PCR) or detection of specific antibodies. The virus neutralization test (VNT) is a gold standard serology test for TAHV diagnosis [7]. Although TAHV infections are common, the disease incidence is underestimated and underreported due to a lack of commercially available diagnostic tests.

In Croatia, the presence of TAHV was confirmed serologically. Very few seroprevalence studies were conducted in the 1970s, showing TAHV antibodies in 7.9% of persons from northeast Croatia and 0.4% of persons from Dalmatia (Adriatic Coast) [17]. The seroprevalence rates among residents of the Croatian islands were reported to be low, ranging from 0.2% (Hvar), 0.34% (Brač) to 1.47% (Mljet) [18]. However, virus circulation was not monitored thereafter. 

Due to the similar seasonal distribution and clinical symptoms (flu-like febrile disease or aseptic meningitis), several other arboviruses could confuse the diagnosis of TAHV infection. Flaviviruses such as tick-borne encephalitis virus (TBEV), West Nile virus (WNV), and Usutu virus (USUV) occur in the continental Croatian regions [19,20,21,22,23]. Although Toscana virus (TOSV) neuroinvasive infections have been reported only sporadically at the Croatian littoral [21], high seroprevalence rates of 53.9% were found among inhabitants of the islands. Seropositive individuals were recorded in the coastal area (33.6%) and mainland of Croatia (6.1%) [24]. Very few clinical cases of Bhanja virus (BHAV) were identified in Croatia in 1970s, including naturally acquired infection presenting with severe meningoencephalitis and spastic quadriparesis, as well as laboratory infections [23]. However, BHAV seroprevalence rates ranging from 11.46% to 58.14% were noted among rural populations on Brač Island [25]. Low seropositivity to BHAV was found on the islands around Zadar (2.2%), Hvar (1.90%), and Mljet (0.1%) [17,18]. In addition to Dalmatia, 7.1% of inhabitants of Northern Croatia tested positive for BHAV antibodies in 1970s [26]. The sandfly fever Naples (SFNV) and Sicilian (SFSV) viruses are widely distributed along the Croatian littoral as well [27].

Clinical cases of human TAHV infection have not been reported in Croatia thus far. The aim of this study was to analyze the prevalence of TAHV antibodies and TAHV RNA in Croatian patients presenting with neuroinvasive disease detected during the arbovirus transmission seasons in 2017–2021.

## 2. Materials and Methods

### 2.1. Patients

The study included 218 patients with unsolved neuroinvasive infection (meningitis, encephalitis, myelitis, ‘febrile headache’) from continental and coastal Croatian regions who developed symptoms during five consecutive arbovirus transmission seasons (April 2017–October 2021). Cerebrospinal fluid (CSF) and serum samples were collected in all patients, while urine samples were available for 90 patients. All samples were collected during the acute phase of the disease. Viral etiology of neuroinvasive infection was suspected based on the CSF findings (pleocytosis, mononuclear predominance, elevated protein level, and normal glucose level).

In the tested group, there were 144 (66.1%) males and 74 (33.9%) females aged from 5 to 91 years (median age 52, IQR = 31–68 years). Patients were from 18/21 Croatian counties (Figure 1A); 172 (78.9%) patients resided in flooded and wetland areas (Figure 1B).

### 2.2. Methods

CSF and urine samples were tested for the neuroinvasive arboviruses TBEV, WNV, USUV, TOSV, TAHV, and BHAV. The presence of viral RNA was tested using real-time RT-qPCR. For TAHV, the following primers and probes were used. Forward primer: 5′-CCATTCCGTTAGGATCTTCTTCCT-3′; reverse primer: 5′-CCTTCCTCTCCGGCTTACG-3′; and probe: FAM-5′-AATGCCGCAAAAGCCAAAGCTGC-3′-TAMRA. In addition, CSF and serum samples were tested for the presence of IgM and/or IgG antibodies (Table 1).

TAHV strain UVE/TAHV/1958/CS/92 grown in Vero E6 cells was used as an antigen for VNT. Virus titers (median tissue culture infectious dose; TCID_50_) were calculated using the Reed and Muench formula. Serum samples were heat-inactivated (30 min/56 °C), and serial two-fold dilutions beginning at 1:5 were prepared. An equal amount (25 µL) of inactivated serum dilutions and 100 TCID_50_ of TAHV were mixed and incubated for one hour at 37 °C with CO_2_. Finally, 50 µL of 2 × 10^5^ Vero E6 cells/mL were added to each well. The plates were incubated at 37 °C with CO_2_ and, starting from the third day, the inoculated cells were inspected daily for the cytopathic effect. NT antibody titer was defined as the reciprocal value of the highest serum dilution that showed at least 50% neutralization. Virus back titration, negative, and low positive control were included in each run. A serum NT antibody titer of ≥10 and a CSF antibody titer of ≥5 were considered positive.

### 2.3. Statistical Analysis

The differences in seropositivity rates according to patients’ demographic/clinical characteristics were compared using a Chi-square test. The strength of association between dependent (VNT positivity) and independent variables was assessed by logistic regression. Statistical analysis was performed using Stata version 16 software.

## 3. Results

Clinical presentations in patients with neuroinvasive disease included meningitis (141/64.7%), meningoencephalitis (56/25.7%), myelitis (8/3.7%), and ‘febrile headache’ (13/5.9%). 

The seroprevalence of detected arboviruses is presented in Table 2. TAHV RNA was not detected in any tested CSF or urine samples; however, TAHV NT antibodies were detected in the serum samples of 22 (10.1%) patients. Two TAHV-seropositive patients (aged 79 and 61 years, respectively) were seropositive for WNV as well, while one (aged 86 years) was seropositive for SFSV.

In two patients aged 59 and 75 years presenting with meningitis, TAHV NT antibodies were detected in the CSF as well. The high NT antibody titers in serum samples (640 and 320, respectively) as well as positive CSF (titers 10 and 5, respectively) suggest recent TAHV infection (Table 3). 

The TAHV seropositivity rates according to patients’ demographic and clinical characteristics are presented in Table 4. The median age of seropositive patients was 63 (interquartile range; IQR = 53–72) years. There was a significant difference in the prevalence of NT antibodies according to age. The seroprevalence increased from 1.8% in patients less than 30 years to 24.4% in patients over 70 years (*p* = 0.001). TAHV seropositivity did not differ significantly between males and females (11.1% vs. 8.1%, *p* = 0.485). In addition, there was no difference in the seropositivity rates among patients presenting with meningitis (8.4%), meningoencephalitis (12.7%), myelitis (14.3%), or ‘febrile headache’ (15.4%). Although the seroprevalence did not differ among inhabitants of suburban/rural and urban areas (10.5% vs. 9.8%, *p* = 0.882), patients residing in floodplains were more often seropositive than patients from other regions (12.2% vs. 2.1%, *p* = 0.044).

The results of the risk analysis show that age was a risk factor for TAHV infection (Table 5). Patients older than 70 years showed a significantly higher risk of seropositivity than those under 30 years of age (OR = 18.064, 95%CI = 2.207–147.809; RR = 13.902; 95%CI = 1.851–104.393). Gender and area of residence were not associated with TAHV seroprevalence. In addition, there was no association between seropositivity and clinical presentation. Living in a floodplain was a risk factor, but did not reach statistical significance.

The NT antibody titers in seropositive patients were 10–640 (geometric mean titer; GMT 86). The median NT titer was 160 (IQR = 20–320). The distribution of antibody titers according to age is presented in Figure 2. The median antibody titer in the age group ≥70 years was 160 years (IQR = 80–320) compared to 40 (IQR = 20–320) in the 50–69-year age group, although the difference was not significant (*p* = 0.782). However, the NT antibody titers in patients older than 50 years were higher than in patients less than 50 years (10–160).

The geographic distribution of TAHV seropositive patients is presented in Figure 3. Patients were from nine Croatian counties, eight continental counties, and one county on the Croatian littoral (Figure 3A,B,D). The majority of patients (20/22; 90.9%) were residents of floodplains along the rivers Drava, Bednja, and Mura (B1), Drava and Danube (B2), Sava (B3), and Dobra (B4). The average altitude at which seropositive patients resided was 128 m above sea level (range 12–323 m). Note that location B5 is on the Adriatic Coast at the mouth of the Rječina River, 12 m above sea level.

## 4. Discussion

Although our seroprevalence studies indicate that TAHV is present in the Croatian population, no human clinical cases have been reported in Croatia to date. Due to mild clinical symptoms in the majority of cases, infections are probably underreported. In the present study, acute neuroinvasive TAHV infections were not confirmed by detecting TAHV RNA in the CSF. However, NT antibodies were detected in the serum samples of 22 (10.1%) patients, indicating exposure to TAHV. Furthermore, the presence of high NT antibody titers in serum (640 and 320, respectively) and detection of NT antibodies in the CSF (titers 10 and 5, respectively) in two patients with meningitis suggested recent TAHV infection.

At 10.1%, the overall seroprevalence among patients with neuroinvasive disease is similar to the seroprevalence in the Croatian general population from north-eastern regions (7.9%) detected in the 1970s [17]. However, previous Croatian studies showed very low seroprevalence rates (0.2–1.47%) among inhabitants of the Croatian littoral [18]. It is important to note that all but one TAHV seropositive patients in this study were residents of the continental Croatian regions.

Epidemiological studies conducted among the general population worldwide have shown significant regional differences in TAHV prevalence. In the 1960s, TAHV infections occurred in most central and southern European countries, and were most common in central Europe, where 30.3–61.9% of the population possess antibodies [35]. Living near rivers and recent flooding events in floodplain ecosystems in Europe appears to be associated with increased seroprevalence in humans [3,14]. Several seroprevalence studies were conducted in the Czech Republic in the 1970s. Seroprevalence rates of 17.8% to 42% were detected in South Moravia [36], 20.6% in the Odra River basin, and 10.1% in the Karvina district. In addition, TAHV NT antibodies were found in 16.6% of forestry employees (an exposed population) from different wooded areas of North Moravia, compared to 5.5% of persons in the control group (inhabitants of Jablunkov and Trinec, an area with a minimum mosquito occurrence) [37]. Very high seropositivity (53.8%) was recorded in the Breclav area, South Moravia, during devastating floods which occurred in 1997 [38]. A seroprevalence study conducted in 1995 among forestry workers in Slovenia, a country bordering Croatia, showed an overall seropositivity rate of 1%. The highest seropositivity was recorded in northern regions (Slovenj Gradec 1.90%, Bled 1.98%, Celje 2.5%) [39]. In the 2000s, NT antibody prevalence was found to be 16.5% in the Czech Republic (rural areas along the Vltava and Labe Rivers affected by the flooding in 2002) [14], 36.3% in Cameroon (rural villages in the south, 2002–2003) [40], and 4.5–18.3% in China (Xinjijang, 2007–2008) [41]. A significantly lower seroprevalence rate (2.0%) was recorded in Iraq (Nasiriyah, 2012–2013) [42]. Epidemiological studies in the Lao PDR indicated high levels of TAHV endemicity in both urban and rural communities. Several studies conducted in the villages of the Nakai plateau showed a TAHV seropositivity of 30.45% in 2007 and 29.06% in 2010 using ELISA [43]. In 2011, TAHV antibodies were recorded for the first time in the Alps. Among 0.3% of VNT positive Tyrolean blood donors, gardening seemed to be a significant risk factor for seropositivity. However, because of the small number of seropositive results and the fact that gardening is a common leisure activity among older people in the Tyrols, it was impossible to identify whether gardening was an independent risk factor or a factor associated with age. Recreational activities such as hiking, cycling, swimming, mushroom harvesting, hunting, and other outdoor activities were not risk factors for TAHV seropositivity [44].

Several studies have tested symptomatic patients for TAHV antibodies. Febrile patients were analyzed in the aforementioned Chinese study (Xinjiang, 2007). TAHV IgG antibodies were detected in 13.0% of tested patients using IFA, while 5.3% were positive for both IgM and IgG antibodies, as confirmed by detecting NT antibodies [41]. Another Chinese study (Geermu, Qinghai-Tibet Plateau, 2009) screened patients with fever from rural clinics who developed symptoms during summer. TAHV IgM antibodies were detected in 2.2% of patients, while 0.9% were positive for TAHV RNA [11]. In 2015, TAHV antibodies were detected in 37.7% of patients under 18 years old who experienced fever and rash in the Lao PDR. In addition, very high seroprevalence rates were detected in patients with dengue and chikungunya fever symptoms, at 92.2% (2013) and 54.68% (2016), respectively [43]. 

Only one study conducted in Russia (Sverdlovsk region, 1994) analyzed the prevalence of TAHV in patients with encephalitis, showing TAHV antibodies in up to 60% of patients [45]. Because of the difficulty in diagnosing TAHV infection due to the lack of commercially available tests, the number of actual cases is likely underreported. Furthermore, patients who do not seek medical treatment or have less severe disease may not be included in case reports.

The results of this study showed age-dependent differences in TAHV seropositivity. A significant increase with age was observed. Seroprevalence ranged from 1.8% in patients under 30 years to 24.4% in patients above 70 years. As in our results, in the Czech Republic the proportion of residents with TAHV antibodies increased with age in disease-endemic areas [14]. The highest positivity rate was found in persons over 59 years (17.53%) compared to children in the age groups 0–5 years and 6–14 years, with seroprevalence rates of 0.00% and 0.56%, respectively [46].

In this study, the majority of seropositive patients were residents of floodplains along rivers, with the average altitude of their residency being 128 m above sea level (range 12–323 m). In the Alps, TAHV infections were reported at 335–1221 (average 497) meters above sea level [44], while in China TAHV was detected in a high latitude region (2800 m above sea level) [12]. In the Czech Republic, the prevalence of TAHV antibodies increased with decreasing distance to floodplain forests, the primary breeding habitat of vector mosquitoes (from 13.6% at distances of >6 km to 28.2% at distances of <1 km) [14]. Seroprevalence was higher in patients from floodplains in this study (12.2% vs. 2.1%). However, living in a floodplain as a potential risk factor did not reach statistical significance in a risk analysis. Due to the small number of patients from other areas, this result should be interpreted with caution. 

Additionally, all Croatian TAHV seropositive patients were from continental counties in which the floodwater mosquitoes *Ae. vexans*, the main vectors of TAHV, are highly prevalent, including Primorje-Gorski Kotar County, located on the Croatian littoral [47,48]. In eastern Croatian regions (flooded areas of the River Drava), *Ae. vexans* is the most abundant species, comprising up to 86% of mosquito populations [49].

Although this study found that males were seropositive more often (11.1%) than females (8.1%), this difference was not significant. Similarly, no association between gender and seroprevalence was reported in the Czech Republic [5]. In the Alpine Central European region of the Tyrol, the majority of seropositive blood donors were males; however, due to a small number of positive individuals, these results should be regarded with caution [44]. Gender differences in TAHV seroprevalence were observed in several provinces in the Lao PDR. While higher seropositivity in males was found in Champasack province in 2013 (67.44% vs. 24.75%) and the capital, Vientiane, in 2016 (30.68% vs. 23.99%), higher seropositivity in females was found in Vientiane province in 2015 (22.96% vs. 14.1%) [43]. 

TAHV-seropositive Croatian patients were detected mainly in the north-western and eastern regions (90.9% of all seropositive). The geographical distribution of seropositive patients overlapped with areas where acute infections caused by other neuroinvasive arboviruses such as TBEV, WNV, and USUV, and seropositive individuals were frequently observed during the same period in Croatia [22,23].

This study has limitations that need to be addressed. Because VNT detects total TAHV immunoglobulins, not antibody classes, the matched serum samples are required to demonstrate an increase in antibody titer to confirm acute infection. For most of the patients included in this study, only one serum and CSF sample were available. However, high NT antibody titers detected in serum as well as detection of NT antibodies in the CSF samples of two patients could indicate a recent neuroinvasive TAHV infection. 

Another limitation is possible cross-reactivity with other California encephalitis serogroup viruses, such as the La Crosse (LACV), Snowshoe Hare (SSHV), Jamestown Canyon (JCV), and Inkoo (INKV) viruses. TAHV is more closely related to LACV and SSHV than to JCV and INKV. Studies in mice have shown that California serogroup viruses induce high levels of homologous response, but different levels of cross-reacting NT antibody response. TAHV has poorer cross-NT antibodies compared to LACV. The highest level of cross-NT antibodies has been observed against SSHV, followed by JCV and then LACV. A lower level of cross-reactivity was detected with INKV [50]. As LACV, SSHV and JCV are all found in the USA, while INKV is primarily found in Scandinavia and Russia, the possibility of cross-reactions with these viruses in Croatia is expected to be low. 

Furthermore, differences in the neutralization test performed may influence the interpretation and comparison of the results. In the patients presented in this study, the TAHV NT antibody titers were from 10 to 640, while GMT was 86 using TCID_50_. A study conducted among the human population in the flooded endemic area of the Czech Republic (2002) titrated positive samples to estimate the dilutions causing plaque-number reduction by 50% and 90% (PRNT_50_ and PRNT_90_, respectively). For PRNT_50_, titers were 32–2048 and GMT 260, compared to PRNT_90_, in which titers were 16–1024 and GMT 119 [14]. 

Moreover, because samples from asymptomatic people were not available, TAHV titers in age and region matched controls were not compared. However, titers in Croatian patients with neuroinvasive disease were compared with titers in asymptomatic people tested in other countries. Using a 100 TCID_50_, as in this study, asymptomatic seropositive residents of a Shanghai suburb had NT antibody titers of 40 to 80 [51]. Furthermore, similar titers of 40 and 60 were detected in an Austrian study conducted among asymptomatic Tyrolean blood donors [44]. In this study, more than half of patients with neuroinvasive disease (54.5%) had NT titers ≥ 160. In addition, NT antibody titers were higher in patients over 50 years of age compared to the patients less than 50 years old. In high-risk areas for TAHV, such as in the Czech Republic, higher titers can be found in asymptomatic individuals [14].

## 5. Conclusions

The presented results demonstrate TAHV activity in Croatia. TAHV seroprevalence is location-dependent, with the highest seropositivity in the north-western and eastern regions. Further research on a larger sample of symptomatic and asymptomatic individuals from various geographic regions is needed in order to assess the prevalence and clinical relevance of this neglected viral disease in the Croatian population.

## Figures and Tables

**Figure 1 microorganisms-10-01443-f001:**
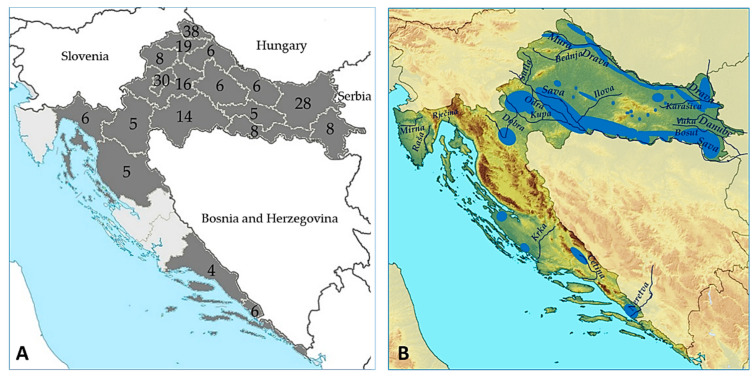
Sampling counties (shadowed gray) and number of patients sampled by county (**A**); Geomorphology of Croatia (**B**). Croatian lowland rivers flow through the plain of the former Pannonian Sea, forming large drainage basins. The longest rivers in continental Croatia are the Sava (562 km), Drava (505 km), Kupa (296 km), and a 188-kilometre section of the Danube. All of them more or less inundate flood meadows and flood forests. Croatia has a remarkable wealth of wetlands (large wetland areas are marked blue). The longest rivers emptying into the Adriatic Sea are the Cetina (101 km) and a 20 km section of the Neretva [28].

**Figure 2 microorganisms-10-01443-f002:**
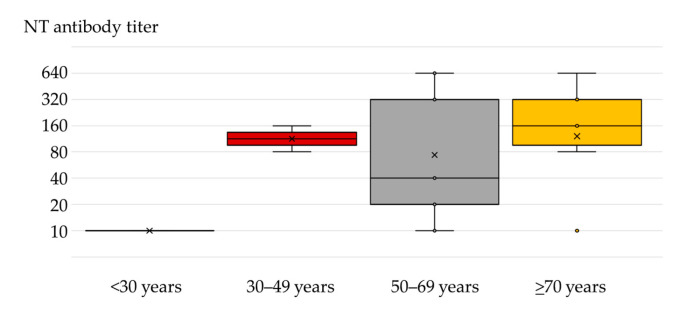
Median (interquartile range) and mean (×) TAHV-neutralizing antibody titers in patients with neuroinvasive disease according to age.

**Figure 3 microorganisms-10-01443-f003:**
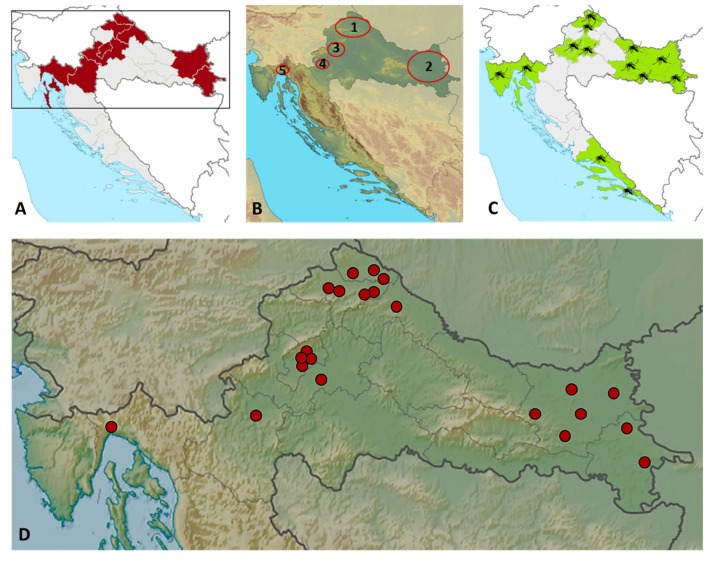
Geographic distribution of TAHV seropositive patients: counties with reported cases are shadowed in dark red (**A**); regions (1–5) with reported cases are circled (**B**); distribution of individual cases by counties (**D**); counties with numerous presence of *Ae. vexans* mosquitoes in Croatia are shadowed in light green (**C**).

**Table 1 microorganisms-10-01443-t001:** Diagnosis of neuroinvasive arboviruses.

Virus	Serology	Manufacturer/Protocol	Sensitivity/Specificity ^a^	RT-qPCR Protocol
TBEV	ELISA IgM/IgG, VNT	ELISA/IFA; Euroimmun, Lübeck, Germany VNT protocol; Ilic et al., 2020 [19]	IgG 100%/100%	Schwaiger and Cassinotti, 2003 [29]
			IgG 100%/100%	
WNV	ELISA IgM/IgG, VNT		IgM 94.4%/99.8%	Tang at al., 2006 [30]
			IgG 99.5%/96.9%	
USUV	ELISA IgG, VNT		100%/99%	Nikolay et al., 2014 [31]
Phlebovirus Mosaic (TOSV, SFSV, SFNV, SFCV)	IFA IgM/IgG		IgM 100%/100% IgG 100%/92%	Weidmann et al., 2008 [32]
TAHV	VNT			Li et al., 2015 [33]
BHAV	NT			Matsuno et al., 2013 [34]

^a^ Sensitivity and specificity stated by manufacturer; TBEV = tick-borne encephalitis virus; WNV = West Nile virus; USUV = Usutu virus; TOSV = Toscana virus; SFSV = sandfly fever Sicilian virus; SFNV = sandfly fever Naples virus; SFCV = sandfly fever Cyprus virus; TAHV = Tahyna virus; BHAV = Bhanja virus; NT = not tested.

**Table 2 microorganisms-10-01443-t002:** Seroprevalence of arboviruses in patients with neuroinvasive disease (N = 218).

Virus	RT-qPCR	IgG Positive N (%)	95%CI
TBEV	Negative	3 (1.4)	0.3–3.9
WNV	Negative	6 (2.8)	1.0–5.9
USUV	Negative	1 (0.5)	0.0–2.5
TOSV	Negative	2 (0.9)	0.1–3.3
SFSV	NT	2 (0.9)	0.1–3.3
TAHV	Negative	22 (10.1)	6.4–14.9
BHAV	Negative	NT	

TBEV = tick-borne encephalitis virus; WNV = West Nile virus; USUV = Usutu virus; TOSV = Toscana virus; SFSV = Sandfly fever Sicilian virus; TAHV = Tahyna virus; BHAV = Bhanja virus; NT = not tested.

**Table 3 microorganisms-10-01443-t003:** Clinical, laboratory, and virology results in two patients with suspected recent Tahyna infection.

Characteristic		Case 1	Case 2	Reference Values
Demographic characteristics	Gender	Male	Female	
	Age	75	59	
Date of disease onset		16 June 2019	18 October 2020	
Area of residence		Urban	Rural, close to a river	
Clinical characteristics	Clinical presentation	Meningitis	Meningitis	
	Clinical symptoms	Fever, headache, weakness	Fever, headache, weakness, nausea, photophobia	
	Underlying diseases	-	Hypertension	
	Outcome	Recovery	Recovery	
CSF findings	Cells (mm^3^)	412	218	<5
	Mononuclear cells (%)	78	70	100%
	Proteins (g/L)	0.7	1.3	0.17–0.37 g/L
	Glucose (mmol/L)	3.1	2.2	2.5–3.3 mmol/L
TAHV RT-qPCR	CSF	Negative	Negative	
	Urine	Negative	Negative	
TAHV VNT (titer)	Serum	320	640	≥10 positive
	CSF	5	10	≥5 positive

CSF = cerebrospinal fluid; TAHV = Tahyna virus; RT-qPCR = reverse transcription-polymerase chain reaction; VNT = virus neutralization test.

**Table 4 microorganisms-10-01443-t004:** Prevalence of TAHV NT antibodies according to patients’ demographic and clinical characteristics.

Characteristic		Tested	TAHV NT Antibodies N (%)	95% CI	*p*
		N (%)			
Gender	Male	144 (66.1)	16 (11.1)	6.5–17.4	0.485
	Female	74 (33.9)	6 (8.1)	3.0–16.8	
Age group	<30 years	57 (26.1)	1 (1.8)	0.0–9.4	0.001
	30–49 years	44 (20.2)	2 (4.6)	0.6–15.5	
	50–69 years	76 (34.9)	9 (11.8)	5.6–21.3	
	≥70 years	41 (18.8)	10 (24.4)	12.4–40.3	
Area of residence	Urban	132 (60.5)	13 (9.8)	5.3–16.2	0.882
	Suburban/rural	86 (39.5)	9 (10.5)	4.9–18.9	
Living in floodplain	Yes No	172 (78.9)	21 (12.2)	7.7–18.1	0.044
		46 (21.1)	1 (2.1)	0.0–11.5	
Clinical presentation	Febrile headache	13 (6.0)	2 (15.4)	1.9–45.4	0.702
	Meningitis	143 (65.6)	12 (8.4)	4.4–14.2	
	Meningoencephalitis	55 (25.2)	7 (12.7)	5.3–24.5	
	Myelitis	7 (3.2)	1 (14.3)	0.4–57.8	

TAHV = Tahyna virus; CI = confidence interval.

**Table 5 microorganisms-10-01443-t005:** Risk analysis for TAHV seropositivity.

Characteristic	OR	95% CI OR	*p*	RR	95% CI RR	*p*
Male (Ref.) vs. female gender	1.375	0.513–3.678	0.525	1.389	0.567–3.401	0.471
Age						
<30 years	Ref.			Ref.		
30–49 years	2.666	0.233–30.400	0.429	2.590	0.242–27.662	0.430
50–69 years	7.522	0.924–61.200	0.059	6.750	0.880–51.770	0.066
>70 years	18.064	2.207–147.809	0.007	13.902	1.851–104.393	0.9010
Suburban/rural (Ref.) vs. urban area of residence	1.069	0.436–2.623	0.882	1.062	0.474–2.377	0.882
Living in floodplain	6.258	0.819–47.820	0.071	5.616	0.775–40.659	0.087
Clinical presentation						
Febrile headache	Ref.			Ref.		
Meningitis	0.503	0.099–2.542	0.406	0.545	0.136–2.179	0.391
Meningoencephalitis	0.802	0.146–4.402	0.799	0.827	0.193–3.528	0.797
Myelitis	0.916	0.068–12.322	0.947	0.928	0.101–8.529	0.947

OR = Odds ratio; CI = confidence interval; RR = Relative risk.

## Data Availability

Not applicable.

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
