# Peer review of "Detection of Tahyna Orthobunyavirus-Neutralizing Antibodies in Patients with Neuroinvasive Disease in Croatia"

_microorganisms, 2022, doi:10.3390/microorganisms10071443_

Round 1

Reviewer 1 Report

The study by Vilibic-Cavlek et al., describes a study of undetermined neuroinvasive disease cases in Croatia and the analysis of primarily serum and CSF samples for the detection of TAHV neutralizing antibodies (as no TAHV viral RNA was detected). The authors make claims about TAHV’s seroprevalence based on these samples and are appropriately careful not to overstate the findings as a diagnosis of TAHV disease. Overall, I think the findings are of interest and relevant, however there are a few concerns that I think should be addressed. The primary concern is that the conclusions are all based on the assumption that neutralizing antibody to TAHV definitively means there was TAHV exposure in those patients, but no other California serogroup virus was tested. The CSG viruses are notoriously cross-reactive, so without at least one other CSG virus comparison it is impossible to conclude that the TAHV NAb are actually TAHV-specific NAb, and not cross-reactive Ab due to some other CSG virus exposure.

Major concerns:

1.     As stated above, there are concerns with concluding TAHV NAb = TAHV exposure. At least one other CSG virus should be included to make the TAHV exposure/seroprevalence conclusion stronger.

2.     The methods describe screening of the samples for additional viruses (TBEV, WNV, USUV, TOSV, and BHAV), but none of the results for those viruses are shown or mentioned. Since testing for these viruses  is mentioned in the methods, the associated results should be included.

Minor concerns:

1.     Introduction page 2, line 47: “In addition” can be removed because the previous sentence focused on the RNA genome, and this sentence focuses on the proteins, so they are not in addition to the genome, they are encoded by the genome

2.     Introduction page 2, line 48-49: The RdRp is part of the L protein, the way the sentence is written makes it sound like they are two entirely separate proteins.

3.     Introduction page 2, lines 55 & 65: Citations should be added for the principal arthropod vector and the asymptomatic human infections.

4.     Introduction: Some added explanation of why neuroinvasive disease cases were looked at when the authors seem to be more generally interested in seroprevalence, which could have been addressed using any serum samples, would help to better frame the paper.

5.     Materials and Methods, page 2, Section 2.1 Patients: Directly state that samples were taken from the acute phase here.

6.     Figure 1: Explanation of the shading and colors is needed. Additionally adding the number of patients sampled per county in Fig 1A would be helpful.

7.     Line 164: Table 2 should be Table 3 in both the text and the Table heading.

8.     Table (3) Risk analysis: Define OR, CI, RR, IQR.

9.     A figure or table showing the individual antibody titers by age group would be more useful than the current figure 2, since really the only associated risk factor was age.

10.  Why wasn’t flood plain included in the risk assessment? This seems like the highest risk factor.

11.  Figure 3: This figure needs a legend to explain all of the colors, circles, numbers, and pictures.

12.  The discussion is overall very good, but could benefit from additional discussion of the caveats with only focusing on neuroinvasive disease patients and the cross-reactivity of other CSG viruses that may confound the results.

Reviewer 2 Report

Vilibic-Cavlek et al. describe a serosurvey of Tahyna (TAHV) virus from patients presenting with neuroinvasive disease in Croatia.  While TAHV is found throughout Europe, it is rarely found to be the causative agent of disease.  This manuscript gives more insight into the seroprevalence of TAHV in Croatia, which may have been the causative agent for severe disease in two cases.  The paper was interesting to read; however, I have a few comments for the authors which may be useful.

Major comments: 

11. Mention of other arboviruses that may be mistaken for TAHV infection are not discussed until the Methods (lines 106-119).  I find it interesting that the authors say, “CSF and urine samples were tested for the most common neuroinvasive arboviruses: TBEV, WNV, USUV, TOSV, TAHV, and BHAV.” I suppose the authors are speaking of the eastern European region?  Even so, is BHAV a common arboviral pathogen?  Only a few cases have been reported, one scientific report in the literature, and the two others in book chapters.  Consider adding a paragraph to the Introduction to familiarize the idea of other neuroinvasive arboviruses in the region that confuse the clinical diagnosis of TAHV infections.

22. While the authors tested the panel of samples for other arboviruses, there is no mention of those test results.  While the study focuses on TAHV seroprevalence, I think it is essential to also report the results of tests with other arboviruses.  Were any samples positive for any other viruses tested?  What was the seroprevalence of the other arboviruses within this sample set?

Minor comments:

11. Introduction (Lines 84-86): The years of the sample collection are only mentioned in the Methods.  Consider mentioning in the abstract as well as the Introduction.

22. Methods (Section 2.1): Please state the protocol number for the Internal Review Board for using human samples for this study.

33.  Methods (Lines 116-119): what are the specificities/sensitivities of the commercial tests used in the study? What is the false negative rate among these serology tests?

  4. Results and Discussion (starting at line 152): In addition to the results of the other serology tests, did any patients with neutralizing antibody to TAHV have positive results for other arboviruses? Consider describing the neutralization test performed in past studies for the detection of TAHV infections and compare it to the TCID50 used in this study.  Are there differences that may alter the interpretation of the results?

Reviewer 3 Report

             This study from Vilibic-Cavlek and colleagues determines the prevalence of Tahyna virus (TAHV) RNA and autoantibodies in 218 Croatian patients with neuroinvasive disease from 18/21 Croatian counties during peak arbovirus transmission seasons; neuroinvasive disease is defined by CSF findings, including pleocytosis and elevated protein levels. Although TAHV RNA was not present in the CSF or urine of these 218 individuals, TAHV neutralizing antibodies were present in 10.1% of patients tested. TAH RNA was detected in the CSF of two out of 75 patients with meningitis. Notably, TAHV seropositivity increased with age and was higher amongst those who lived near the floodplains. Seroprevalence of TAHV is 60-80% in endemic countries, including the Czech Republic; previous studies have reported a 7.9% prevalence of TAHV seropositivity in northeast Croatia, 0.4% in Dalmatia, and between 0.2-1.4% in the Croatian islands. These data demonstrate that TAHV is present in Croatia and provides new information on neutralizing antibody titers in individuals with neuroinvasive disease from that country. 

Major comment:

            This report would have been improved by including TAHV titers in age/region matched controls without neuroinvasive disease and by including longitudinal sampling from patients.  But, overall, an important snapshot of neutralizing antibody responses in patients with neuroinvasive disease for this understudied virus. 

Minor comments:

1.      Line 51, quotation mark needs to be raised before “Valtice fever”

Round 2

Reviewer 1 Report

All of the concerns have been adequately addressed. 

Author Response

Dear Editor,

Informed Consent Statement has been added (lines 385-386).

Best regards,

Tatjana Vilibic-Cavlek and Vladimir Stevanovic

Reviewer 2 Report

The authors have addressed all comments from the original review except with regard to patients' consent to use their samples for research purposes.  Did the authors receive permission from the patients to use their samples in this study?  This should be explicitly stated in the methods.

Author Response

(The authors gave the same response as above.)
